# *Homo sapiens* May Incorporate Daily Acute Cycles of “Conditioning–Deconditioning” to Maintain Musculoskeletal Integrity: Need to Integrate with Biological Clocks and Circadian Rhythm Mediators

**DOI:** 10.3390/ijms23179949

**Published:** 2022-09-01

**Authors:** David A. Hart, Ronald F. Zernicke, Nigel G. Shrive

**Affiliations:** 1Department of Surgery, University of Calgary, Calgary, AB T2N 4N1, Canada; 2McCaig Institute for Bone & Joint Health Research, University of Calgary, Calgary, AB T2N 4N1, Canada; 3Faculty of Kinesiology, University of Calgary, Calgary, AB T2N 1N4, Canada; 4Bone & Joint Health Strategic Clinical Network, Alberta Health Services, Edmonton, AB T5J 3E4, Canada; 5Department of Orthopaedic Surgery, University of Michigan, Ann Arbor, MI 48109-5328, USA; 6School of Kinesiology, University of Michigan, Ann Arbor, MI 48108-1048, USA; 7Department of Biomedical Engineering, University of Michigan, Ann Arbor, MI 48109-2099, USA; 8Department of Civil Engineering, Schulich School of Engineering, University of Calgary, Calgary, AB T2N 4V8, Canada

**Keywords:** acute conditioning, acute deconditioning, myokines, osteokines, ground reaction forces, connective tissue homeostasis, circadian rhythms

## Abstract

Human evolution required adaptation to the boundary conditions of Earth, including 1 g gravity. The bipedal mobility of *Homo sapiens* in that gravitational field causes ground reaction force (GRF) loading of their lower extremities, influencing the integrity of the tissues of those extremities. However, humans usually experience such loading during the day and then a period of relative unloading at night. Many studies have indicated that loading of tissues and cells of the musculoskeletal (MSK) system can inhibit their responses to biological mediators such as cytokines and growth factors. Such findings raise the possibility that humans use such cycles of acute conditioning and deconditioning of the cells and tissues of the MSK system to elaborate critical mediators and responsiveness in parallel with these cycles, particularly involving GRF loading. However, humans also experience circadian rhythms with the levels of a number of mediators influenced by day/night cycles, as well as various levels of biological clocks. Thus, if responsiveness to MSK-generated mediators also occurs during the unloaded part of the daily cycle, that response must be integrated with circadian variations as well. Furthermore, it is also possible that responsiveness to circadian rhythm mediators may be regulated by MSK tissue loading. This review will examine evidence for the above scenario and postulate how interactions could be both regulated and studied, and how extension of the acute cycles biased towards deconditioning could lead to loss of tissue integrity.

## 1. Introduction

### 1.1. The Purpose of This Review

Tissues of the musculoskeletal system (MSK) are designed to function within the boundary conditions of Earth and, thus, are influenced by variables such as gravity, the geomagnetic field, and other variables such as oxygen tension and temperature. MSK tissues require the biology of their cells to function in mechanically active environments. As such, MSK tissues all function in the context of a “use it or lose it” paradigm and require mechanical loading to maintain their integrity. While it is convenient to study each tissue of the MSK system in isolation, it is clear that the results of such studies do not reflect the complexity of their regulation from a systems biology perspective where their regulation is integrated with that of other biological systems.

Therefore, the purpose of this review is to present evidence that maintaining the integrity of MSK tissues requires not only regular bouts of mechanical activation but also periods of mechanical inactivity during which growth and repair activities occur via biological signals. Such cycles of mechanical activity–inactivity would likely correlate with circadian rhythms and thus, would have to be integrated with the mediators of the 24 h circadian cycles on Earth. Thus, this review brings together lines of investigation to generate a perspective on how to approach new directions for research in MSK tissue repair and how to address diseases of these tissues. Furthermore, validation of the concept may also identify some of the reasons why progress in developing effective interventions in diseases that impact such tissues has been slow. For example, large numbers of studies in this area use nocturnal rodent models, but they are studied by human researchers who work on the animals during the active phase of the human circadian cycle! Thus, better understanding of the concepts raised could not only provide new insights into the regulation of tissues of the MSK system, but also how more informative research could be conducted.

### 1.2. Background

During evolution leading to *Homo sapiens*, with their upright walking and navigation via mobility through the environment, hominoids experienced loading of their lower extremities during walking and running, either barefoot or with their feet covered in a minimalist covering to protect the feet from injury. Thus, the loadbearing connective tissues were exposed to various forms of biomechanical stimuli, ground reaction forces (GRFs), muscle contractions, and for intra-articular tissues of a joint such as the knee, intermittent hydrostatic compression due to loading of the synovial fluid as well as shear via relative motion of the cartilage and menisci surfaces and tension on the ligaments of the knee. Thus, during the day, the cells and tissues of the lower extremities (i.e., bone, muscles, tendons, and intra-articular tissues such as cartilage, menisci, ligaments, synovium, and capsule) and the spine are loaded repeatedly. In contrast, during periods of prone rest, some loading of various tissues may occur due to leg and back motions and influence of the 1 g environment, but there is minimal to no exposure to GRFs. Thus, during a typical 24 h cycle of a day, aspects of the body would be subjected to distinct periods of varying biomechanical input. An example that such periods of varying input can lead to alterations in connective tissues is findings that compression of the spinal intervertebral discs (IVD) due to bipedal loading during the day is released during the night, and most young humans are taller in the morning than at night.

It is also clear that humans are subjected to well characterized circadian rhythms that are reflected in day–night cycles (Figure 1), but nearly all of the focus has been on the biology of the circadian rhythms, and biomechanical influences on such rhythms and their integrity and effectiveness have only been considered peripherally [1,2,3]. Two environments investigated over the past few decades, space flight and a surrogate for space flight, chronic 6-degree-tilt head-down bedrest, indicate that removal of a human from the ground reaction forces caused by our 1 g environment leads to deconditioning of multiple physiological systems including bone and muscle of the lower extremities [4]. In the case of bedrest, deconditioning occurs rapidly in the presence of the 1 g background of Earth, while that of space flight occurs in the absence of the 1 g background.

Those findings suggest that previously not identified cycles of acute conditioning and deconditioning may be intrinsic to normal human physiological responses and function. While circadian rhythm patterns are well known and often studied in the context of drugs and medication efficacy [6,7,8], and tissue repair and regeneration [9], the primary focus of extant studies has been on biological responses (e.g., based on a search of the PubMed database). However, during 8 h of uninterrupted sleep in a bed, a human is also subjected to a substantial period without the usual effects of weightbearing ground reaction forces. Granted that throughout evolution 8 h of uninterrupted sleep in a bed was not a standard for humanoids but sleep in a prone position for a variable period of time did occur.

Likely an even more feasible approach to understanding the potential of acute biomechanical-based conditioning–deconditioning occurring in human programming could come from fetal development, the early post-natal period of growth, and then the post-puberty years to skeletal maturity. During fetal development, a human is suspended in an aqueous environment that is exposed to gravity, but the fetus is not directly exposed to ground reaction forces (GRFs). Thus, the cardiovascular system (CVS) develops in the absence of an environmental regulator, as do the tissues of the musculoskeletal (MSK) system. While the conditioning of the cardiovascular system via the pumping of blood and the associated shear stresses remains intact, there are no ground reaction forces to provide a conditioning stimulus for the tissues of the MSK. After joint cavitation there can be movement of the limbs, but in the absence of GRFs. 

After birth, during early post-natal growth, the neonate may experience some GRFs via crawling, but it is not until the initiation of walking and subsequent navigation that a human experiences GRFs in a consistent manner. This GRF exposure is interrupted by extended periods of sleep (8–10 h) with associated removal of GRFs. As the onset of walking can vary from ~7 to ~15 months of age, the pre-walking programing is variable and such programing may include developing the ability to incorporate GRFs into physiologic systems without injury. Post-walking, the young human still sleeps in the prone position, which is a potentially deconditioning condition. Whether growth occurs mainly during such a deconditioning environment or irrespective of such a consideration is not well known. It is possible that conditioning stimuli during exposure to continual, but intermittent, GRFs may be counter to growth so removal of the GRF vector may be needed for growth-associated activity, a concept consistent with the perspective of Rogers et al. [9]. Potentially, the circulating anabolic factors in a young human require the availability of functional receptors to affect growth, and that process could be interfered with by GRFs. Following the onset of puberty the same principles would hold, with the major change being the types and concentrations of the circulating mediators. For females, the levels of the sex hormones as mediators of the circulatory and MSK systems also undergo additional cyclic variation (i.e., menstrual cycles) that would also have to be accounted for during both conditioning and deconditioning.

After skeletal maturity, potential cycles of acute conditioning–deconditioning events may be critical for maintenance of the system with the decline in the levels of circulating anabolic stimuli associated with growth and maturation. That may be why physical activity and programmed conditioning via exercise are critical for system integrity as levels of anabolic mediators become more pronounced. This may be why physical activity and exercise are effective as a preventive measure when it comes to loss of system integrity during aging, as it can be considered a surrogate anabolic response or serves as an adjunct to exercise-dependent mediators such as myokines and other mediators released by exercise [4].

With aging, the ability to maintain the integrity of the MSK systems, for example, becomes more difficult due to both challenges in continuing physical activity above a necessary threshold and altered sleep patterns, as well as genetic and epigenetic variables that were not apparent as a young person when other variables were sufficient to control them. Interestingly, space flight, which Ray [10] postulated represented “accelerated aging,” is often accompanied by disrupted sleep patterns in astronauts and disrupted circadian rhythms [11,12].

Thus, acute periods of engaging the GRFs of the Earth followed by periods of non-engagement (i.e., “deconditioning”) may play important roles at defined periods across the lifespan of humans. By extension, such a regulatory process may also be involved in maintaining the integrity of tissues of other land-dwelling species because the basic processes involved at the molecular and physiologic levels would have preceded the evolution of bipedal *Homo sapiens* and their predecessors.

## 2. The Potential Role of Mediators and Receptor Expression Generated by Physical Activity and Engagement of GRFs to Set the Stage for Subsequent Effector Processes

If the above hypothesis regarding a system of acute cyclical patterns of “conditioning–deconditioning” involving GRFs is a mechanism used by humans for regulating the integrity of physiological systems, the question arises as to how physical activity involving GRFs could mediate the effects. Likely, there is more than one possibility and some of the options are not mutually exclusive. The first is via increased heart rate induced by physical activity/exercise, either aerobic and/or resistance, but mobility associated activities would likely engage GRFs more than resistance exercises. The second possibility relates to the engagement of muscles in the activity and the release of myokines, which are known to exert effects on a variety of target tissues, including the cardiovascular system and brain structures [4]. Therefore, muscle is a secretory organ [13]. The myokines released by muscles may depend on the type of physical activity [4]. While physical activity can facilitate maintenance of the muscle system during aging, certainly a number of people develop sarcopenia during aging, and that can be exacerbated in those with obesity [14,15,16]. Myokines released from activated muscles include irisin, interleukin-6 (IL-6), brain-derived neurotrophic factor (BDNF), myostatin, and fibroblast growth factor-2 (FGF2) (Figure 2) and can lead to muscle–bone crosstalk [17]. In addition, the mechano growth factor, also known as IGF-1E, is a splice variant of IGF-1 that is expressed by activated muscles [18,19]. Chang et al. [20] have reported that irisin levels may have predictive value for sarcopenia. Furthermore, Kim et al. [21] have reported that resistance exercise can lead to increases in irisin in both aging humans and mice. Thus, myokines have the potential to be generated during physical activity and could subsequently impact target tissues, including bone [22,23,24,25], both during the activity, as well as during the acute deconditioning part of the day while asleep or at rest.

In addition to myokines, there are also osteokines released from bone when subjected to exercise [28]. Interestingly, osteokine expression after exercise involving jumping is decreased in post-menopausal females compared to younger women (Figure 2). 

As GRFs can stimulate both muscles and bone, those two components of the musculoskeletal system function as a “muscle–bone” unit [29,30]. In addition, there appears to be considerable “crosstalk” between bone and muscle with myokines and osteokines central to such communication [17,31,32,33,34]. Both osteokines and myokines may also have a role to play in bone loss during space flight [35]. Interestingly, in a preclinical rat model, treadmill running (with GRFs) and swimming (without GRFs) influenced cardiovascular (CVS) adaptations differently [36]. Whether such differences are also observed in humans remains to be defined. Therefore, mobility and navigation through the environment or via exercises involving GRFs may be required to optimally condition the CVS, as well as other physiological systems via cycles of acute conditioning with GRFs and then acute deconditioning via bedrest during a wake–sleep cycle. 

In summary, exercise and loading of muscle and bone can result in the release of biologically active molecules that can influence tissues and cells of the MSK system, as well as other organ systems. Whether such molecules act immediately after release in an autocrine, paracrine, or endocrine fashion, or require a period of rest or minimal loading such as at night during the dark period of the circadian rhythm remains to be determined.

## 3. The Need to Integrate Activity-Generated Mediators with Circadian Rhythm Variations during Periods of Acute “Deconditioning” 

### 3.1. Background

If a day is a mix of activity-based exposure to GRFs and then a period of inactivity or acute “deconditioning”, and this aligns with the day/night cycle, then night-time is not just a period of time that consists only of inactivity. Humans experience daily circadian rhythms [37,38,39,40,41] but are somewhat heterogeneous with regard to the details of the circadian rhythm patterns they experience [42]. Furthermore, if the periods of time, particularly during growth and maturation, were not restricted to the day/night paradigm but to a series of bouts of inactivity during the day, such inactivity periods during the day would not be identical to those at night due to variations in the levels of molecules regulated by the circadian processes. Thus, circadian rhythms would potentially impose complexity on the environment at night due to the presence of mediators at different times, including an early morning increase in corticosteroids, which are known to influence the cells of a number of tissues [43,44]. Such variation may have relevance to MSK tissue healing as Dietric-Zagonel et al. [45] recently reported that the steroid dexamethasone enhances Achilles tendon healing in a rat model, but this enhancement is dependent on the timing of drug administration.

Furthermore, levels of growth hormone (GH) also vary during the day, with the highest levels occurring before skeletal maturity and at night during the period of “deconditioning” from the perspective of engagement with GRFs [46]. In addition, sex differences in growth hormone (GH) secretion patterns are also known for males and females during the lifespan [46,47]. Thus, two important conclusions can be drawn from the GH story, (1) the highest levels of the hormone are at night when relative inactivity or deconditioning occurs and (2) any mediators resulting from engagement between tissues such as muscles and bone with GRFs must interact with a varying environment of biologically active mediators under the control of the circadian rhythm cycle. In humans, such diurnal variations lead to elevated levels of molecules released from a number of brain structures including the pineal glands, suprachiasmatic nucleus (SCN), hippocampus, and hypothalamus [37,38,39,40,41]. These include melatonin, which reaches high levels at night (Figure 1) and is a molecule that has been reported to exhibit anabolic influences on bone both in vitro and in vivo [48,49,50,51,52], as well as a number of other relevant tissues and cells (Table 1). Interestingly, the in vitro studies have been performed in the absence of artificial loading of the cells, and therefore, it is not yet apparent whether loading in vitro would modulate the responsiveness of the cells to melatonin. That could be readily assessed using a system to apply artificial loads during exposure to melatonin.

Furthermore, melatonin levels can decrease with age [52], can be disrupted by shift work [68], and are reported to be disrupted during time in space [69,70,71]. How disruption of circadian rhythms contributes to changes associated with microgravity has not been explored in any detail. However, this could be addressed, in part, by taking supplemental melatonin in a time-dependent manner and with various doses as postulated by Ikegame et al. [72]. Initially, supplemental melatonin was taken as a sleep aid at night Chase and Gidal, 1997 [73], but it has also been taken for other applications Marqueze et al., 2021 [74], and some adverse events have been noted Foley and Steel, 2019 [75]. 

As discussed above, GH is another very relevant mediator of tissue growth and repair that exhibits a circadian rhythm pattern of expression [46], and as such is highly relevant to the concept of alternating acute periods of activity/inactivity being important for regulating the responsiveness of MSK tissues to mediators derived from activity, as well as circadian rhythms. Interestingly, in rats subjected to the hind limb suspension model of disuse, GH was found not to stimulate recovery of bone formation but did positively influence muscle growth [76]. Thus, in this circumstance, disuse perhaps led to a failure to express functional GH receptors in bone, and the GH supplemented rats did not respond to the hormone. That study supports the conclusion that loading via GRFs leads to hormone receptor expression, similar to what was observed in the rabbits with an immobilized leg [77].

While the above perspective is only a hypothesis presently, it is interesting that most research has focused only or mainly on the biology of circadian patterns of activities. The potential role of biomechanics in regulating the maintenance of many physiological systems or whether such biomechanical stimuli have been incorporated into biological programing has been largely unstudied. Thus, the importance of GRFs and the associated physical and exercise activities as a conditioning stimulus across the lifespan remains a productive area for future research.

There is a complicating scenario in the above hypothesis that is related to results from the study of bedrest participants. As noted, during prolonged bedrest there is bone loss, and some studies indicate that the osteopenic responses can begin acutely within 1–2 days [78]. However, as reported by Custard et al. [79], night-time levels of melatonin were still elevated in a group of male bedrest participants over 17 days. Therefore, if melatonin is an anabolic mediator for bone, why do chronic bedrest participants still lose bone? Without further information, it is not possible to advance a detailed answer to that question, but there are some possibilities that are testable. The first is that melatonin does not work in isolation and requires additional factors derived from exposure to GRFs. A second possibility is that GRFs work via a direct impact on osteocytes or other bone cells to enhance expression of functional melatonin receptors (i.e., MT1, MT2, or both) [80,81,82], and thus, to achieve a melatonin effect expression of functional receptors to facilitate signal transduction to the cell interior is required. A third possibility is that bedrest deconditioning leads to the expression of an inhibitor that blocks the ability of melatonin to stimulate the bone cells, and expression of this paracrine inhibitor is blocked by exposure to GRFs. Lastly, the findings from prolonged bedrest studies may provide another possibility to explain the disconnect between melatonin levels and the bone loss observed. Those findings raise the possibility that exposure to GRFs also leads to the expression of functional expression of receptors for a diverse set of molecules, and with bedrest and a lack of GRF exposure, functional receptors for melatonin are not expressed, and thus, the melatonin is present but cannot exert a biological response. That this could occur comes from in vivo evidence from the study of the growth of knee ligaments in very young rabbits with one leg rigidly immobilized [83,84,85,86]. These findings are discussed in more detail later in this review.

These options are only examples of approaches to answer the question posed, and the answer(s) may not only support the prevention of bone loss during bedrest, but also in space. In addition, the conundrum related to regulation of bone loss in the upper extremities in either bedrest or during space flight still needs explanation.

### 3.2. Biological Clocks, Circadian Rhythms, and Regulation of MSK Tissue Homeostasis and Repair 

As reviewed by Dudek and Meng [87] and Yang and Meng [88], circadian rhythms are regulated by circadian clocks that are both central in nature and relate to clock genes expressed in brain centers, as well as peripheral in nature and regulated by clock genes in a tissue-specific manner. Thus, tissues of the MSK system can be regulated by light/dark central regulators, as well as those regulated at the individual tissue level. Since articular cartilage is devoid of vascularity and innervation, this implies that the intrinsic circadian rhythm of cartilage is not directly influenced by neural elements as discussed in [87,88,89]. Both the tissue-specific and central clocks operate on a 24 h cycle. As discussed in [87,88,89], disruption of clock genes centrally or in tissues of the MSK system such as cartilage can lead to a catabolic state and risk for osteoarthritis development. Knocking out one of the essential clock genes of chondrocytes, *Bmal1*, leads to loss of cartilage homeostasis and tissue integrity in mice [90]. Biological clocks can regulate inflammasome pathways [91] and the disruption of the circadian clock can be induced by inflammation [89]. Furthermore, such clocks can also be affected and entrained by exercise [87,88,92]. Some of the effects of biological clocks and circadian rhythms on relevant MSK tissues and cells are outlined in Table 2 to assist the reader with interest in specific tissues.

Of relevance to the topic at hand is the finding that exercise can entrain the biological clocks of tissues of the MSK system and, thus, there is integration of the central and tissue-specific peripheral circadian rhythms and exercise, or biomechanical loading is an important component of that potential integration. That leads to the question of “what is the best time to exercise in order to achieve the best integration and perhaps, the best outcomes?” [106,107,108]. As presented by Brito et al. [106], evening exercise for humans, prior to the rest/inactivity part of the day, may offer better outcomes than morning exercise. It is also relevant to point out that exercise can lead to release of biologically active myokines and osteokines, which can influence a number of cells of the MSK system as well as others. This has been discussed recently [35,109]. Thus, if such exercise-induced mediators have a defined half-life, then generating increased levels during a time immediately preceding a rest or “deconditioning” period of time could not only lead to enhanced responsiveness, but also require more integration with circadian-rhythm-dependent molecules such as melatonin.

It is also well known that rats and mice can be genetically selected for high and low spontaneous wheel running during the active part of their daily circadian rhythm [110,111], which is at night as they are nocturnal. Thus, the question then arises as to whether the entrainment of the biological clocks of their central and peripheral tissues is influenced by such differences in exercise levels. It is interesting that a high-fat diet [112] can interfere with such wheel running, but there appear to be sex differences in the responses in mice [113] with female mice more resistant to the effects of the diet. The basis for such sex differences is not well defined, but melatonin levels are reported to be influenced by the menstrual cycle and menopause in humans [114]. This type of diet can also influence biological clocks [115,116,117]. Recently, it was reported that exercising rats while on a high-fat high-sucrose diet completely eliminated the biological consequences of that diet on tissues of the male MSK system but only if started at the same time as the diet [118,119]. Of note was that the treadmill aerobic exercise in this model was performed during the day and thus was performed during what would have been the resting part of the circadian rhythm for these rats. Theoretically, such exercise should have perhaps been disruptive to the circadian rhythm and biological clocks of these rats. Additional studies with such rats may provide further insights into factors regulating the integrity of the biological clocks, exercise-dependent mediators and responsiveness to such obesity-inducing diets. As disruption of circadian rhythms and biological clock integrity by shift work can influence health at multiple levels [120,121], such studies could have an impact at multiple levels.

From the above discussion, it is clear that effective integration of mechanical loading of tissues of the MSK system and its consequences with the elements of the circadian rhythms and biological clocks is essential for maintaining the integrity of these tissues of the MSK system, and that coordinated periods of activity and inactivity may play an essential role in such regulation (Figure 3).

## 4. Indirect Evidence for Conditioning–Deconditioning in Regulation of Connective Tissue Growth and Maintenance

Several decades ago, studies by Rubin et al. indicated that a minimal number of loading cycles was sufficient to maintain the integrity of bone in the turkey ulna [122,123]. Thus, continual “conditioning” was not required to maintain integrity. One interpretation of such findings is that release of myokines and osteokines from a minimal number of cycles of loading was all that was required for subsequent regulation of bone cells. Which cells released such mediators is not clear, but potentially they could be from osteocytes which are believed to be the “mechano-sensing” cells in bone [124,125].

From the work of Frost [126,127,128,129], the mechanostat hypothesis for the regulation of bone in the 1 g environment of Earth appears to be a viable approach to understanding the regulation of bone during growth and maturation, as well as maintenance when skeletally mature [129]. The application of Wolff’s Law to explain variations in bone structure in lower and upper extremities is well supported by the literature. Thus, bone is very adaptable to the mechanical environment. However, the lower extremities are exposed to GRFs, while the bones of the upper extremities are exposed to muscle loading in the context of the 1 g environment, but with no GRF loading. Interestingly, astronauts in space lose more bone in the lower extremities than the upper [130,131], even though they typically use their upper extremities to propel themselves in the 0 g environment. This differential bone loss suggests that bones in the lower extremities are more acutely affected by factors including exposure to GRF loading than those in the upper extremities.

Nevertheless, assessments of bone densities in athletes involved in different sports indicate that bones of the upper extremities can adapt to impact loading. For example, the bones of the forearm of tennis players exhibit increased density only in the forearm used in the sport [132]. Athletes involved in jumping sports such as volleyball have lower-extremity bone densities elevated compared to swimmers or cyclists [133] and, hence, exposure to GRF loading can impact bone structure. Thus, as predicted by the work of Frost [126,127,128], the bone adapts to meet the demands within a physiologic window. Overuse or fatigue without sufficient time or circumstance to allow for repair can lead to fractures or loss of integrity [129]. How exactly the GRF loading is converted to bone adaptation is not known in detail as it could relate to a direct effect of such impact loading on the cells in the tissues of the lower extremities, due to the release of mediators (i.e., osteokines or myokines) from the cells affected, or a combination of both. If the response to GRFs were a direct effect on the cells in tissues of the lower extremities, the GRF effect would likely decrease from the foot/ankle to the proximal hip as one might expect a gradient of adaptation as one progresses up the motion segments of the lower extremity. One might also predict that bone density of the femur of a lower extremity fitted with an artificial below-the-knee prosthesis after an amputation would be different than that of the contralateral femur, if the loading was a direct effect on cells of the bone and related tissues even if some aspects of loading were similar [134,135,136]. The lower extremity motion segment also has two sites of discontinuity, the ankle and knee, and thus the GRF loading must traverse these two sites to reach the femur. Interestingly, the bone densities of sheep that walked on a level surface or via a 15% incline for 45 days were not different indicating that the angle of the knee discontinuity did not overtly influence the bone density in that time span [136]. Therefore, such findings may indicate that the response to GRFs may involve systemic or local mediators released from the more distal aspects of the lower extremities. However, this would not explain the differences in density of lower vs. upper extremity bones, or the above-discussed findings that the racket-held forearm of a tennis player used in the sport was larger and more dense than in the contralateral non-racket limb. Some evidence for the role of mediators in the regulations of bone under “deconditioning” conditions comes from reports such as that of Colaianni et al. [137], who reported that the mediator irisin, a myokine, can prevent bone loss in a mouse hindlimb suspension model.

In preclinical models, it has also been shown that immobilization of a rat lower extremity leads to muscle atrophy, but results in a longer femur than that of the contralateral leg [138]. Thus, immobilization can lead to dysregulation of growth control in rodents.

Similarly, in a rabbit model, it has been shown that immobilizing one hind limb in maximal flection using a rigid K-wire at 3 months of age leads to a cessation of ligament growth when assessed after 1 month of immobilization [83,84]. It was confirmed that the animals did not use the immobilized limb based on force plate analysis but were as active as non-immobilized age-matched controls. However, the contralateral knee ligaments were not normal biomechanically, so some limb crosstalk was noted. For example, the laxity, stiffness, and stress at failure of the medial collateral ligament (MCL) contralateral to the experimental MCL were different from the control values [86]. Whether such differences were due to overuse or due to neural signaling via the dorsal root ganglion is not presently known. In this circumstance, the bones of both hind limbs were not detectably different [85].

In contrast, when the rabbit knee was immobilized when the animals were 4 weeks of age, a different outcome was obtained [77,86]. The bones of the lower extremity of the immobilized limb continued to grow during immobilization and eventually grew past the joint line, and the joints could not be opened after removal of the K-wire. Thus, in the very young animal, the anabolic factors present during that period of rapid growth continued in the absence of GRFs. The contralateral limb was “normal” regarding the bone alignment in the knee so there was no collateral influence on bone growth, but there was a contralateral effect regarding the ligaments of the knee [77,86]. Interestingly, similar treatment of rabbits at 6 weeks of age did not lead to the same severity of joint growth abnormality, but the ligaments were still compromised in both the experimental and contralateral legs [86]. As young rabbits gain 72% of their adult mass and 94% of their adult tibial length by 4 months of age [139,140], most of that early tibial bone growth can be associated with the growth plate of the lower leg, and thus, chronic unloading of the tibia can lead to excess longitudinal bone growth. By extension, that implies that GRF loading of the tibia is a regulating factor in bone growth in this early post-natal growth. One potential mechanism is that loading modulates the responsiveness of cells in the growth plate to anabolic factors. The fact that bone growth in the tibia of the loaded contralateral leg is “normal” at the macro level indicates that the changes with immobilization are unique to the affected limb, and not mediated by neural factors at the level of the Dorsal Root Ganglion of the spine.

These rabbit studies suggest that: (1) lower limb bone growth is likely the driver for limb growth early in life, and other tissues of the knee adapt in response to the bone growth; (2) loss of regulation of the process by removing GRF loading leads to abnormal growth; (3) GRFs and loading are required to maintain coordinated development of a functional knee; (4) by 3 months of age, this coordinated limb development is established, and immobilization no longer leads to abnormal further development, possibly due to a decline or loss of anabolic mediators; and (5) loading of the leg leads to modulation of growth of the tibia, likely mainly at the level of the growth plate. Of importance to the present discussion is that conditioning of the bone during the early phase of life interferes with the activity of these anabolic factors. Unfortunately, the quality of the bone from the immobilized limbs, the contralateral limb, or age-matched controls was not assessed to determine whether the bone exhibited inferior qualities compared to those from animals proceeding through maturation normally. Based on Frost’s work [126,127,128,129], one would expect that the conditioning of the growing bone via GRFs would contribute to the development of the template for the subsequent development of a mechanostat. 

The question that remains from the above-noted studies is whether bone growth and development of the mechanostat occurs during the loading (conditioning phase) or during the off-loading phase (at night/during “rest”, deconditioning). However, the above studies and the earlier discussion do indicate that bone is a unique responder to GRFs which is essential for the integrity of lower extremity bones. Of note is the fact that astronauts lose more bone from their lower extremities than the upper extremities [141], which is confirmation that loss of GRFs and the 1 g environment is of critical importance, and that merely exercising via motion is not sufficient to maintain optimal bone integrity [142].

For effective bone formation and adaptation, combining loading with intermittent rest (unloading) of various intervals has been shown to stimulate bone formation. The beneficial effects of load–rest cycles on bone have been demonstrated for preclinical models [143,144,145], growing children [146], and middle-aged and older individuals [147]. To provide a theoretical context for these effects, Gross and colleagues [142] synthesized the data suggesting that allowing bone to rest between loading cycles may transform “… low- and moderate-magnitude mechanical loading into a signal that potently induces bone accretion”. They hypothesized that the osteogenic nature of rest-inserted loading arises by enabling osteocytes to communicate as a small world network [142].

## 5. Why Might Cycles of Acute “Conditioning–Deconditioning” Be Important in Regulation of Lower Extremity Connective Tissues?

As noted above, during development and early post-natal life the fetus is shielded from ground reaction forces (GRFs) during a period of rapid and complex growth. Subsequently, after walking the young human is subjected to GRFs via cyclic periods of mobility and rest. The periodicity of such cycles may reflect the need to generate mediators during periods of activity, which then require a period of time in the absence of GRFs to influence target cells and tissues.

An example of how biomechanical stimulation can influence the activity of a mediator on cells or tissues is derived from studies of IL-1 effects on chondrocytes [148], macrophages [149], cartilage [150,151], or menisci [152]. The interaction of IL-1 on these cells was inhibited if the cells or tissues were subjected to biomechanical loading. This may also indicate that in the joint that has excess IL-1, the effect of the interleukin would be more influential during a sleep period than during the day when exposed to GRFs associated with mobility. While it has not been explored in detail, such inhibition may also apply to other mediators including anabolic mediators present during periods of growth and maturation. These periods require sensitive regulation to achieve the necessary coordination, for instance, that which is required to have legs of equal length and joints with the appropriate tissues functioning as an organ system [153,154]. For this paradigm to work, the “deconditioning” that would occur at night must also interact with molecules/mediators of circadian rhythms, a set of circumstances that may also exert some regulatory influences on the impact of cytokines and growth factors on the appropriate cells.

After skeletal maturity, when levels of anabolic mediators decline (except after an injury or fracture when there is a transient expression of anabolic mediators contributing to fracture healing and injury repair), such cycles may also adapt to prevent deconditioning from adversely affecting the integrity of the tissues. A myriad of studies suggest that connective tissues subscribe to the paradigm of “use it or lose it”, and muscle hypertrophy and atrophy, as well as immobilization effects on bone are examples. From studies with tissues from preclinical models, it has been shown that removal of a connective tissue such as a knee meniscus from its in vivo mechanical environment leads to the de-repression of a number of catabolic mediators by 4 h, and that this de-repression can be prevented by exposing the tissue to cyclic hydrostatic compression [155]. Thus, maintenance of exposure to conditioning stimuli after skeletal maturity and during the aging process becomes critical to offset such potential catabolic de-repression.

Clearly, the cells in tissues such as a subset of those of the knee would still be loaded somewhat during prone rest cycles via motion of the legs and muscle activation while prone, but they would lack direct exposure to GRFs. Thus, there may be some unique aspects of responsiveness to GRFs for tissues of the lower extremities that are currently not appreciated to their full extent. Some of the uniqueness may reside in the release of specific myokines and osteokines from tissues exposed to GRFs within a physiological window that allows for optimal release of the appropriate mediators. Such mediators could then exert their effects via autocrine, paracrine, or endocrine mechanisms on physiological systems expressing receptors for the myokines and osteokines. That approach could be evaluated using serum biomarker level assessments before, during, and after treatments involving exposure to a controlled GRF.

## 6. Are Responses to GRFs Mechanistically Unique Due to Exposure to Them Late in Evolution?

It is commonly suggested that “life” began as single-celled organisms in the oceans with its salt and mineral content. Once cells acquired mitochondria and developed the ability to reproduce, further evolution to multicellular organisms occurred. During that time, cells likely developed the need to counteract compressive forces at various depths of the ocean, as well as to resist shear forces after attaching themselves to rocks or other surfaces resulting from wave motion and ocean currents. Eons later, marine life appears to have evolved into reproducing functional multicellular organisms with differentiated functions. During that time, both internal and external accommodation of compressive and shear stresses were likely optimized within the boundary conditions of their environments. At some point, creatures ventured out of the oceans and onto land, becoming air breathers and being exposed to GRFs. Exposure to the 1 g on Earth via GRFs required either adaptation of existing mechanisms to respond to compression and shear, or development of new paradigms to adjust to surviving on land. Thus, the evolution of adaptive appendages such as legs and forearms was necessary (except for species such as snakes). Whatever adaptations were required to enhance functionality and survivability to breathing oxygen and walking on land obviously could not interfere with the functioning of mechanical mechanisms to respond to compression and shear that were ingrained into single-cell mechanisms.

Before progressing too far via evolution, exposure to GRFs by the initial organisms that ventured out of the water onto land would require adaptation not only of the bony skeleton and muscles, but also the regulatory elements of bones and muscles, as well as the supporting tissues required for mobility via articulating joints (i.e., ligaments, tendons, cartilage, menisci, synovium, and joint capsules) and a segmented spine. Of particular importance would have been the evolution of the “bone–muscle units” with extensive crosstalk [23,156,157], as well as the nerves and vasculature that regulate such tissues. While the vasculature would have been subjected to shear while living and evolving in the oceans and lakes, with the advent of walking on land the vasculature of the lower extremities would also have been exposed to GRFs along with the bones and muscles, and as such, would be required to adapt to such impact loading without initiating damage. Interestingly, such adaptations may eventually start to fail in some people during aging given the large number of people with Peripheral Artery Disease (PAD) in their legs [158,159]. 

Considerable effort has been expended towards understanding the role of GRFs on bone regulation directly or indirectly via muscles [160,161,162]. However, how those GRFs were distributed would likely influence the biological response pattern. For organisms that walked or were mobile via four appendages, GRFs would occur in all four limbs with differences between fore and hind limbs according to weight distribution, e.g., [163]. Except for some birds, kangaroos, and humans, four limbs interacting with the ground appears to be the predominant response pattern that was optimized for each species. That was not always the case as some dinosaurs apparently used their hind legs mainly for mobility. Based on other primates, it is also likely that precursors of *Homo sapiens* also used four limbs for mobility but at some point, became bipedal, which offered some survival advantage. Thus, human arms were subjected to GRFs during the early part of evolution but did not depend on GRFs for function since humans became bipedal. Certainly, the muscles and bones of human forearms can still adapt to impact loading, which at least partially mimics GRFs as evidenced by the arm used by tennis players [132]. Thus, as humans evolved, GRFs were mainly felt via the lower extremities during walking and running to, for instance, secure food and escape predators. 

For a GRF to have an influence on the connective tissues of the leg through to the spine, the loading parameter has to traverse the bones of the leg (i.e., ankle, tibia/fibula, femur, pelvis, and segmented spine). Thus, there are “discontinuities” that the forces would have to traverse at the ankle, knee, hip, lower spine, and between IVD in the spine if the GRF is to have a direct influence on the response mechanism(s) as one moves from the point of impact to the height of the human—granted initially humans were quite short. With the GRFs initiated at the foot/heel, such discontinuities would potentially lead to loss of loading intensity for bones via joints where the bones were interrupted by soft tissues and fluids. Thus, the force gradient from the point of impact would drop considerably by the time the loading response impulse is transmitted to the femur, particularly if the ankle and knee were in flexion and if the loading magnitude drops below a threshold, thereby neutralizing any direct influence. 

From the work of Frost [126,127,128,129] and others [10,164], it is clear that other mechanical forces on the femur can lead to alterations in bone density around the hip due to forces associated with loading deformation contributing to the “mechanostat” or set point of the bone in specific locations. However, such deformation loading may be more influenced by muscles and local environments than GRFs generated at the foot. In the tibia, calf muscles can exert significant loading on the bones to result in changes (part of the muscle–bone unit) [23,156,162]. Whether such muscle–bone interactions in response to GRFs persist above the knee remains to be determined. 

An alternate hypothesis to avoid the discontinuity conundrum and to avoid interfering with prior mechanistic commitment to compression and shear loading is one that involves the nervous system as the vehicle to transmit the loading initiated by the GRFs to bones and muscles of the whole motion segment that constitute the lower extremities. As there are nerves in the lower extremity that traverse the complete length of the leg, it is possible that GRF loading leads to a signal in such nerves that is transmitted up the leg to the dorsal root ganglion (DRG) and back down to the individual tissues (e.g., bones and muscles) to affect the intensity, duration, and frequency of the signal. During times of no GRFs, the neural component of that regulatory system would be neutral, thus, allowing for other mediators to exert their influences on the tissues. The existence of such a neuro-centric system could be assessed by stimulating only one leg with GRFs and assessing responses in the other (i.e., assuming the signals go to the DRG and are then able to affect both limbs, or not). Similarly, exposure of a limb affected by a stroke to GRFs should not lead to contralateral effects. Finally, exposing the lower limbs of spinal-cord-injured individuals with injuries to specific levels to GRFs should not evoke a response, but the bones should still be impacted, and thus, impairment of the neural component to the transmission of the loading signal may or may not affect the outcomes. Exposure to microgravity via space flight or as a result of prolonged bed rest would lead to decreased GRFs, disruption of the muscle–bone unit integrity directly, and loss of the neural input needed to maintain integrity via intermittent exposure to GRFs.

## 7. Further Testing the Acute “Conditioning/Deconditioning” Hypothesis

The hypothesis presented could be directly tested using equipment that supplies a surrogate for GRFs via an impactor to the heel either in space (in the absence of a 1 g influence) or with bedrest (while prone but in the presence of 1 g). The effectiveness of such an intervention could also be assessed as a preventative measure or as a recovery intervention after the loss of bone and muscle from baseline or compared to the contralateral leg. A device that delivers an impact loading comparable to a heel strike has been tested on the MIR space station and shown to be effective in preventing bone loss [165].

Further development and refinement of such equipment would need to be regulated to administer a defined GRF-equivalent with variable frequency and intensity per application, and variable applications per day or per week. Equipment may also need to be adapted specifically for astronauts in microgravity vs. those on Earth and being subjected to chronic bedrest as they are somewhat different environments with different complicating variables. In addition, study participants of differing ages could also be assessed to determine potential aging effects on bone responsiveness in the context of melatonin levels. Thus, investigation of this hypothesis may provide some basic insights into connective tissue regulation by GRFs, as well as develop into a potential intervention with effectiveness to prevent or inhibit alterations associated with space flight and prolonged bedrest. Although that approach would provide information about the use of a GRF analogue for a chronic condition, the findings would not relate directly to the acute situation or the proposal that acute bouts of conditioning/deconditioning occurred normally and are important for maintaining the integrity of the bone–muscle units.

Further studies could be envisioned with workers performing shift work in jobs requiring activities that involve exposure to GRF loading and those that do not (i.e., computer work at a desk). Evaluation of systemic mediators (i.e., myokines and osteokines) in serum from those working during the day vs. night could be evaluated to gain insights into interactions with circadian rhythm mediators regarding levels. Evidence for this concept has been reported by James et al. [166] and Ritonja et al. [167], indicating that shift work can lead to alterations in circadian rhythm mediator expression as well as epigenetic alteration to relevant genes. For others, it may even be possible to perform such studies using individuals who work alternating shift work, and thus potentially addressing individual variation in responsiveness. That such shift-work-associated alterations may lead to risks to the health of these individuals has been discussed by Boivin et al. [121]. Thus, in shift work there would be alterations to the timing of loading of MSK tissues and alterations to the circadian rhythms, potentially leading to the opportunity for dysregulation of the MSK tissues.

In addition to the actual physical labor and exposure to GRFs associated with shift work, two additional factors associated with shift work could also both influence circadian rhythm integrity and the consequences. These two factors are an abnormal exposure to light, Stevens and Zhu, 2015 [121,168] and the timing of food ingestion/eating, Page, 2021; Basolo et al., 2021 and Sinturel et al., 2022 [169,170,171]. During evolution, exposure to light correlated with the day–night cycle, but now humans are exposed to artificial light at night. Similarly, the circadian rhythm is involved in lipid metabolism Sinturel et al., 2022 [171], and eating food at night can influence metabolism, contributing to development of obesity Li et al., 2020 and Kawai, 2022 [172,173], and obesity can further influence circadian rhythm regulation Grosshans et al., 2016 [174]. Thus, the effect of obesity on osteoarthritis could be due to a variety of factors such as increased stress on tissues, while including disruption of circadian rhythm regulation [104]. 

Thus, these preceding, potential study directions have been focused more on a set of chronic models than the initial concept of acute patterns of “Conditioning/Deconditioning” having regulatory importance in lower limb homeostasis. Nevertheless, in the adult, the outcome of such studies could likely facilitate pursuit of experiments to assess the acute aspects of this concept.

## 8. Author Opinion

The above discussion provides evidence from a variety of diverse fields highlighting how biomechanical and biological inputs can significantly affect the regulation of tissues of the MSK system. In the opinion of the authors, this review provides a solid basis for the regulation of these tissues through alternating acute cycles of mechanical activity related to the functioning of the tissues of the MSK system with periods of relative mechanical inactivity. That cyclic pattern allows for the tissues to respond optimally to biological signals, which can facilitate both growth and maturation of the tissues and repair of microdamage. Thus, the regulation of MSK system tissues is influenced by the fact that the tissues operate in mechanically active environments to facilitate mobility as well as being subject to biological stimuli; they must respond according to biologically active principles. Such regulation would be compatible with the well-known 24 h circadian rhythm of humans and undoubtedly is well-integrated into that circadian cycle to operate optimally.

Further characterization of this concept may provide new insights into the regulation of tissues of the mechanically active MSK system, as well as impact thinking regarding the models being used to study such regulation of disease processes and development of the timing of new interventions to restore damaged MSK tissues.

## 9. Conclusions

From the above discussion, it is clear that prolonged periods of disuse, with the absence of exposure to GRFs, can be detrimental to the integrity of many MSK tissues. In humans, such tissues are exposed to GRF loading of the lower extremities for periods of time during the day and then periods of time when they are not exposed to GRF loading. During evolution, the latter likely occurred at night and thus corresponded with aspects of the circadian rhythms of humans. Evidence in the literature indicates that loading of cells in tissues of the MSK system inhibits the responsiveness of such cells to mediators such as growth factors or cytokines, and thus responsiveness may require that the cells be in a more unloaded environment to exert their biological activities. This may be relevant to both times of growth and development, as well as after skeletal maturity to maintain tissue integrity. Furthermore, as injured tissues of the MSK system are most often immobilized after injury (i.e., casting of a broken bone, torn tendon or ligament), there is likely a need to re-introduce gradual increases in loading to both strengthen the repair tissue and re-establish a normal pattern of “conditioning–deconditioning”. Thus, for injuries to MSK tissues of the lower extremities, it may be important to re-expose them to GRFs as soon as appropriate in a manner that recapitulates a normal cycle. The timing of such loading/unloading activities must integrate with the mediators of the circadian rhythms of humans to exert fine regulation (Figure 1). Details of how such integration is manifested remain to be elucidated. While throughout most of evolution such day/night patterns of loading and unloading with regard to GRFs were dominant, more recent work schedules with shift work could lead to disruption of the integrity of such a regulated system and contribute to risk for disease and compromise of elements of the MSK system. In addition, most humans today wear shoes which have changed the pattern of loading to our lower limbs [175], so how the lower extremities are loaded and the intensity of the GRF loading are also factors that need to be considered as the research moves forward in this area. Furthermore, advanced aging with its greater propensity for frequent and prolonged periods of unloading could also contribute to loss of MSK integrity. Collectively, these topics hold promise for generating significant new and in-depth insights with well-designed research studies going forward. As circadian rhythm cycles appear to affect the activities of some cancer cells [176], it is thus likely that mediators of such cycles also affect normal cells of the MSK system, and the interactions with these uniquely mechanically loaded cells and tissues is influenced by that mechanical loading. Finally in any new studies, the variability of responsiveness to stimuli and interaction of mediators as a function of the genetic make-up of the individuals in the study group must also be considered.

## Figures and Tables

**Figure 1 ijms-23-09949-f001:**
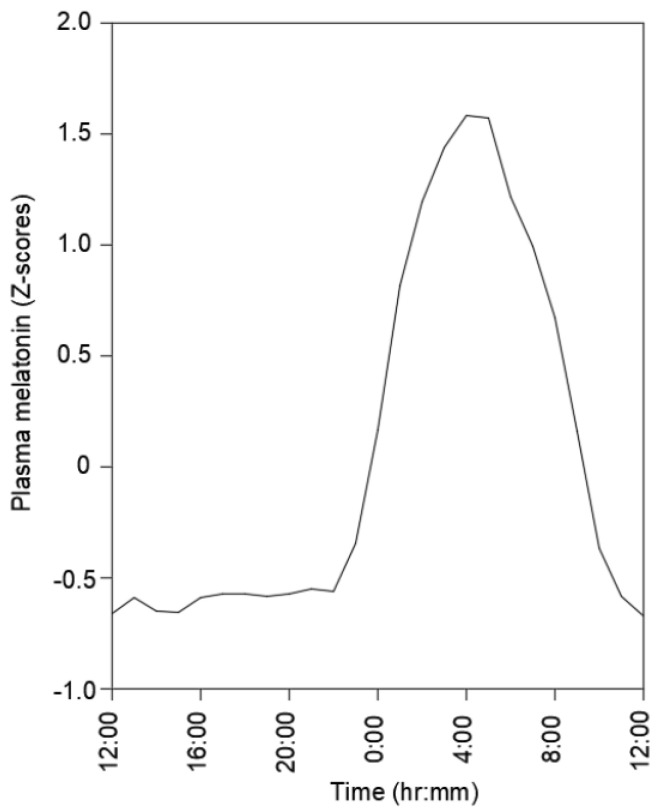
Melatonin variation in plasma over a 24 h period—drawn from data reported in Dijk et al. [5] for seven 21–25 year old males, presented as Z-scores to reduce variability.

**Figure 2 ijms-23-09949-f002:**
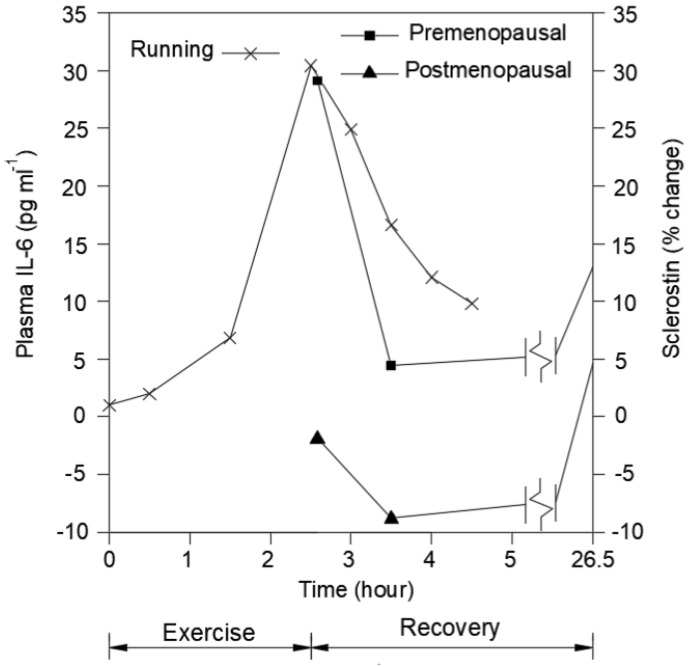
Data for the cytokine IL-6 as reported in Steensberg et al. [26] are plotted (Running data); the data were obtained from seven male endurance trained runners aged 24–50 years who ran on a treadmill at 75% VO2 max for 2.5 h, having reported to the lab at 0800. The sclerostin (osteokine) data (solid squares and diamonds) are—as reported in Nelson et al. [27]—acquired from 20 pre-menopausal and 20 post-menopausal women who arrived at the lab between 0800 and 0900, were provided a specified breakfast and then did a series of jumping exercises before blood samples were taken. Baseline data for these latter two groups are zero on that scale.

**Figure 3 ijms-23-09949-f003:**
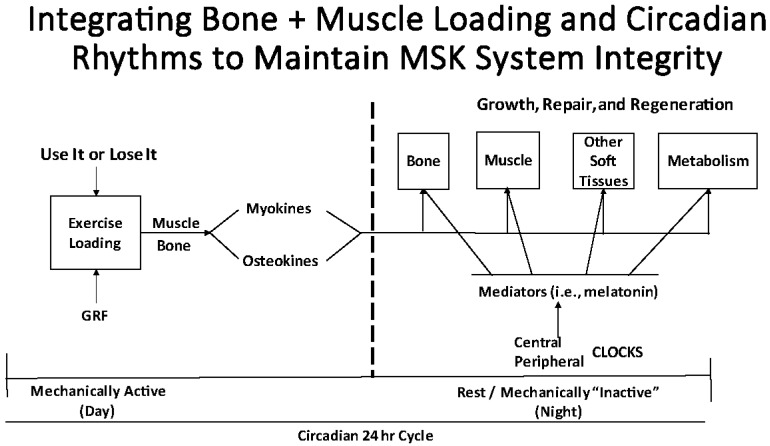
Schematic of the potential effects of loading on the MSK system and how this might require integration of exercise/loading-associated mediators (myokines and osteokines) with mediators following a circadian pattern.

**Table 1 ijms-23-09949-t001:** Influence of melatonin on MSK tissues, cells, and repair.

Tissue/Condition	Species	Type	Year	Citation
Skeletal Muscle	General	Review	2020	Chen et al. [53]
Bone	General	Review	2021	Lu et al. [54]
	Mice	Original	2012	Histing et al. [55]
	Rats	Original	2020	Kose et al. [56]
	Rats	Original	2015	Arabaci et al. [57]
Cartilage	Pig	Original	2009	Pei et al. [58]
Tendon	Rat	Original	2019	Song et al. [59]
	Rat	Original	2021	Zhang et al. [60]
	Rat	Original	2022	Yao et al. [61]
Adult Stem Cells	Human	Original	2014	Liu et al. [62]
	General	Review	2014	Luchetti et al. [63]
	General	Review	2017	Zhang et al. [64]
	Human	Original	2014	Lee et al. [65]
	Human	Original	2014	Gao et al. [66]
Wound Healing	Human/Mice	Original	2014	Lee et al. [65]
Osteoarthritis	General	Review	2021	Lu et al. [67]

References cited are representative of the literature and additional references can be found in the cited publications.

**Table 2 ijms-23-09949-t002:** Influence of circadian clocks and rhythms on MSK tissues and tissue repair.

Tissue/Condition	Species	Type	Year	Citation
Corneal Repair	Mice	Original	2017	Xue et al. [93]
Skin Wounds	General	Review	2022	Fawcett et al. [94]
Tissue Regeneration	General	Review	2021	Ruby et al. [95]
Tissue Homeostasis	General	Review	2014	Janich et al. [96]
Fracture Healing	Mice	Original	2016	Kunimoto et al. [97]
Bone Adaptation	Mice	Original	2022	Bouchard et al. [98]
Bone Turnover	Rats	Original	2022	Song et al. [99]
Bisphosphonates	Rats	Original	2003	Shao et al. [100]
Bisphosphonates	Patients	Clinical Trial	2008	Generali et al. [101]
Adult Stem Cells	General	Review	2014	Brown [102]
	General	Review	2014	Janich et al. [96]
Embryonic Stem Cells	Human	Original	2022	Naven et al. [103]
High Fat Diet/OA	Mice	Original	2015	Kc et al. [104]
Post-Traumatic OA	Rats	Original	2022	Song et al. [99]
IVD Degeneration	Rats	Original	2021	Ding et al. [105]

OA = Osteoarthritis; IVD = Intervertebral Disc; Original = Original Research. References cited are representative of the field and additional references can be found in the cited materials.

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
