# Peer review of "Homo sapiens May Incorporate Daily Acute Cycles of “Conditioning–Deconditioning” to Maintain Musculoskeletal Integrity: Need to Integrate with Biological Clocks and Circadian Rhythm Mediators"

_ijms, 2022, doi:10.3390/ijms23179949_

Round 1

Reviewer 1 Report

The authors present an interesting concept which is important for the understanding of musculoskeletal tissues and provides impact on therapeutic approaches to improve healing and repair. They discuss the cross-talk between mechanosensation, particularly the effect of growth reaction forces (GRF) and circadian rhythm regulation. The developmental situation is represented. The describe how conditioning and deconditioning phases coordinate regular bone growth during development. This review is well written and the concepts conclusive. I have only minor comments.

The effect of obesity – so far known - on the regulatory processes could be addressed.

Page 3 (last paragraph): the abbreviations (Il-6 (write capital letter “L”), BDNF, FGF2) should be explained. The mechanogrowth factor (MGF, a splice variant of IGF-I) could be mentioned here.

Formatting on page 5 the surplus blank in the reference brackets should be deleted (first and last paragraph).

End of the first paragraph: please write “Achilles tendon”

Last paragraph: a point ist lecking before “Interestingly”

The flex cell system does not provide a homogenous loading in contrast to other more appropriate systems, hence it should not be pointed out here. Simply, leave this more common.

Table 1: where the studies cited conducted with or without loading?

Page 6: “taking supplemental melatonin” dosis? Time point of application?

Page 7, third paragraph: “acutely” how rapidly? Hours or days?

“…indications indicate…” style

“That this occur…discussed below” better to cite already here the reference

Page 11, first paragraph: the sentence “As the impact loading… involved” is too long, please rewrite for a better understanding.

“in preclinical models,…” not sure whether an “on” “the contralateral site” is lacking. “Adult” Rodents have an open growth plate.

Third paragraph: “were not normal biomechanically” what differs in detail? “contralateral effect regarding the ligaments of the knee” which effect? It is known that the joint capsule contacts after longer periods lacking joint movements.

Page 12, first paragraph: “by 3 month of age” there is no upright wait at this age only crawling

Third paragraph: “positively” could be simply omitted.

Last paragraph: “interleukin” could be abbreviated as before “IL”

Page 13: “except after an injury..” this should be explained

Third paragraph: “segmented spine” does it mean the spine without the sacral bone?

Reference list: there are some formatting inconsistencies

Reviewer 2 Report

The manuscript addresses an interesting topic but the present organization of the manuscript lacks reader interest. In my opinion, the following suggestions may be helpful in further improvement.  

1. It will be more interesting to modify the present form of abstract as structured abstract. Content and flow of abstract should be modify as per: (background,/objective of the review; methodology; results and finding reported in the literature; conclusion/author opinion).

2. Highlight the current research gap and future directions in this area which the authors want to direct through the literature evidence in the introduction section.

3. Table 1 and 2 should include a column and highlight the outcome of the cited study.

4. Some interesting figures should be incorporated to highlight the significance of the topic for easy understanding to the reader and increase readership on this topic.  Some images from the previous research study which is related to this topic should be incorporated into the revised manuscript. It will be interesting for the reader. 

5. It is highly recommended to include a new section as "author opinion" before the conclusion. The present conclusion section should be a little concise and report only the key message. 
